# Prevalence of Antibiotic Residues in Pork in Kenya and the Potential of Using Gross Pathological Lesions as a Risk-Based Approach to Predict Residues in Meat

**DOI:** 10.3390/antibiotics12030492

**Published:** 2023-03-01

**Authors:** Nicholas Bor, Alessandro Seguino, Derrick Noah Sentamu, Dorcas Chepyatich, James M. Akoko, Patrick Muinde, Lian F. Thomas

**Affiliations:** 1International Livestock Research Institute (ILRI), Nairobi P.O. Box 30709-00100, Kenya; 2The Royal (Dick) School of Veterinary Studies, University of Edinburgh, Easter Bush Campus, Midlothian EH25 9RG, UK; 3Faculty of Veterinary Medicine, University of Nairobi, Kangemi P.O. Box 29053-00625, Kenya; 4World Animal Protection, Nairobi P.O. Box 66580-00800, Kenya; 5Institute of Infection, Veterinary and Ecological Sciences, University of Liverpool, Leahurst Campus, Neston CH64 7TE, UK

**Keywords:** antibiotic residues, food safety, gross pathological lesions, maximum residue limits, public health

## Abstract

The human population is growing and urbanising. These factors are driving the demand for animal-sourced proteins. The rising demand is favouring livestock intensification, a process that frequently relies on antibiotics for growth promotion, treatment and prevention of diseases. Antibiotic use in livestock production requires strict adherence to the recommended withdrawal periods. In Kenya, the risk of residues in meat is particularly high due to lack of legislation requiring testing for antibiotic residues in meat destined for the local market. We examined pig carcasses for gross pathological lesions and collected pork samples for antibiotic residue testing. Our aim was to determine if a risk-based approach to residue surveillance may be adopted by looking for an association between lesions and presence of residues. In total, 387 pork samples were tested for antibiotic residues using the Premi^®^Test micro-inhibition kit. The prevalence of antibiotic residues was 41.26% (95% CI, 34.53–48.45%). A logistic regression model found no significant associations between gross pathological lesions and the presence of antibiotic residues. We recommend that the regulating authorities strongly consider routine testing of carcasses for antibiotic residues to protect meat consumers. Future studies should research on farming practices contributing to the high prevalence of residues.

## 1. Introduction

The global human population has been steadily rising, and estimates project an increase from the current 7 billion to 9.6 billion people by 2050. Most of the population growth is expected in Africa, which has an annual growth rate of 1.2% [1]. The ballooning population, urbanisation and improved incomes are likely to increase the demand for animal-sourced protein in low-income countries [2]. To cope with this increasing demand, livestock keepers are likely to use more antibiotics to prevent diseases and promote growth [3]. 

Globally, it was reported in 2017 that about 73% of all antimicrobials produced are used within food animal production [4] with pigs receiving the highest amount of antimicrobials at 172 mg/kg produced [5]. It is predicted that antibiotic use will rise by 67% between 2010 and 2030 [5] and usage is likely to be relatively higher in sub-Saharan Africa than other regions due to higher disease burden, limited diagnostic capacity, strained health facilities, limited personnel training on antibiotic use and unregulated access to antimicrobials [6]. Tetracyclines, sulphonamides, aminoglycosides and beta-lactams are the most commonly used classes of antibiotics for treatment and prevention of infections in food animals in Kenya with no current evidence of their use specifically for growth promotion [7]. 

Although antibiotics have the potential to improve production levels with regard to producer income and food security, when used inappropriately residues can be present in meat. These residues present public health risks such as allergenicity, toxicity, carcinogenicity and disruption of normal gastrointestinal flora therefore becoming a One Health issue. Residues may also promote the development of antibiotic-resistant microbes [8]. There have been concerted efforts by European Union (EU) member countries to reduce antimicrobial use in food animals by regulating veterinary antibiotic use and medicated feeds [9]. 

Failure to observe withdrawal periods has been cited as a major contributor to antibiotic residues in foods of animal origin [10]. Prior research has indicated that livestock farmers in Kenya have relatively unrestricted access to many antimicrobials without any prescription [11] despite having limited knowledge on withdrawal periods. A study in Busia County, Kenya, for example, showed that 12.9% of the interviewed farmers had no knowledge on withdrawal periods while 34.3% had scanty information on withdrawal periods, indicating a very high potential for antibiotic residues to be present in the meat produced by this group of farmers [12].

Ideally, food meant for human consumption should be free of antibiotic residues. Since a zero-residue level in meat may be impractical, the joint Food and Agriculture Organization of the United Nations and World Health Organization (FAO/WHO) Codex Alimentarius Commission has set maximum residue limits (MRL), above which meat is considered unsafe for human consumption [13]. This recommendation has led to routine national surveillance of antibiotic residues in meat products in most European countries. In Denmark for example, 0.1% of slaughtered pigs are routinely sampled and tested for antibiotic residues [14]. However, the situation is different in most African countries where financial constraints are widespread, and weak regulatory frameworks as is the case in Kenya [15]. As a result, national surveillance of residues remains a challenge [16]. 

In Switzerland, studies have established that risk-based sampling is 100% efficient in detecting tetracycline residues in calves [17]. Another risk-based surveillance study in Denmark found that chronic pleuritis may be used as a risk indicator for carcasses with antibiotic residues [18]. Although these risk-based approaches to monitoring and surveillance to antibiotic residues have the potential to be economically efficient, it is yet to be demonstrated in the Kenyan context [19].

To date, and to our knowledge, no study has investigated the prevalence of antibiotic residues in pork in Kenya. Therefore, this study aimed to establish the prevalence of antibiotic residues in pigs slaughtered and consumed in Nairobi and its environs. As a preliminary step to considering a risk-based surveillance system for Kenya, we also investigated the association between gross lesions detectable at slaughter and antibiotic residues in order to determine if gross lesions detectable at slaughter may be a suitable indicator to use in such a program. We hypothesized that animals with gross pathological lesions may have either been treated with antibiotics resulting in residues or may have originated from farms with generally poor husbandry practices, increasing the likelihood of both poor adherence to withdrawal times and presence of gross lesions in pigs presented for slaughter. 

## 2. Results

A total of three hundred and eighty seven pork samples, each from an individual pig, were collected and tested for antibiotic residues. 126 of these 387 meat samples tested positive for the presence of antibiotics residues above the MRL. The apparent prevalence of antimicrobial residues above the MRL in the sampled population was 32.55% (95% CI 28.08–37.37%). Considering the reported sensitivity and specificity of the diagnostic assay, a true prevalence of 41.26% (95% CI 34.53–48.45%) was calculated. The prevalence of gross lesions in the full study population has been previously described [20] and only data from the 387 pigs with samples available for testing was analysed and reported in this paper. Table 1 shows the univariate analysis results for association of variables with presence of antibiotic residues at 5% significance level. 

On univariate analysis, no statistically significant associations were found between gross pathological lesions and antibiotic residues in pork samples as seen in the last column in Table 1.

Sex, liveweight, tail bite lesions and hindlimb bursitis variables had *p* values of <0.2 and thus were included in the logistic regression model. Variable combination with the lowest Akaike Information Criterion (AIC) figure was picked as the best model. This model included sex, tail bites and hindlimb bursitis variables. The variance inflation factors output for sex, tail bites and hindlimb bursitis were 1.005, 1.008 and 1.01, respectively, indicating that these predictor variables were unrelated. No statistically significant associations were identified from the logistic regression model as shown in Table 2. 

## 3. Discussion

We identified a high prevalence of antibiotic residues above the MRL in pork consumed in Nairobi and its environs. These findings indicate poor adherence to withdrawal periods by farmers and potential public health hazards to pork consumers through toxicity, allergic reactions and potential contribution to antimicrobial resistance development. Contrary to our hypothesis, we found no statistically significant associations between gross lesions and antibiotic residues in this population. 

Due to resource limitations, we utilised Premi^®^Test (R-Biopharm AG, Pfungstadt, Germany) a broad-spectrum antibiotic screening method to detect the presence of residues above the MRL. Although the Premi^®^Test has been certified for use as a screening test based upon its comparability to current reference tests [21], the sensitivity and specificity are still suboptimal at 72.5% sensitivity and 95.3% specificity. Within statutory residue surveillance programs, positive screening results are generally followed by a confirmatory test to avoid false positives [22]. This indicates the potential for both false negatives and false positives when using the Premi^®^Test as a sole analytical technique, as was used in this study. 

Although we adjusted the reported prevalence to account for the reported diagnostic performance, the sensitivity and specificity of the Premi^®^Test can vary among antibiotic classes. This means that there is a possibility that certain classes of antibiotic were not detected in this study. Sub-optimal detection of tetracyclines [23,24] and quinolones [25] in chicken by the Premi^®^Test have previously been reported, indicating that a parallel screening test sensitive for these classes of antibiotic in particular may be required to ensure all positive samples are accurately detected. 

Freezing has been demonstrated to reduce the concentration of some antibiotics in meat [21] which may have resulted in an artificially reduced prevalence of samples with residues above the MRL in our study. The effect does, however, appear to be time related and we therefore expect that immediate freezing of the samples at −80 °C for under six months with a single defrosting event will have reduced the impact of freezing to an acceptable level [26]. More research is required to quantify what impact this may have on residue collection and is highly relevant for the design of future studies or proposed surveillance activities.

Despite the possibility that some positive samples may have been incorrectly classified as negative through use of an imperfect screening test and freezing effects, there is a worryingly high prevalence of residues over MRL as indicated in this study. One possible explanation for the high prevalence of antibiotic residues in pork is that Kenyan authorities lack laws requiring testing of antibiotic residues for locally consumed meat [15]. 

The lack of current legislation in this area results in no financial or legal consequences for farmers and traders to present animals for slaughter where residues may still be present. Lack of legislation and current paucity of data on residues in food products may also lead to low level of awareness among farmers and consumers on the potential public health threats of these residues perpetuating a relaxed attitude to withdrawal periods. 

It has been established that 100% of veterinary shops in Nairobi sell antibiotics without prescriptions with the buyer’s/or farmer’s preferences guiding purchasing decisions [11]. In rural areas of western Kenya, 60% of veterinary shop attendants have been reported to sell antibiotics to farmers without asking for a prescription [12]. This suggests that antibiotic accessibility is higher for pigs raised in urban areas than rural areas, which is potentially due to higher demand for animal-sourced proteins among urban residents with better incomes. Antibiotic residues have also been observed in Kenyan milk [27], indicating that the problem of antimicrobial residues should be addressed across different sectors for a safer food system.

The current legislative deficit can be rectified by regulatory bodies, such as the Kenya Veterinary Board (KVB) and the Veterinary Medicine Directorate (VMD). The two bodies should impose stricter regulations on sale of antibiotics by ensuring that antimicrobials are only accessible to licenced animal health practitioners. This should be combined with raising awareness among stakeholders regarding the negative health impacts of antibiotic residues in animal-sourced proteins. 

Kenya’s National Action Plan on containment of antimicrobial resistance proposes residue testing as a strategic intervention [28]. The establishment of such a residue surveillance program would require investment in diagnostic technologies (e.g., mass spectrometry for confirmatory testing), training of staff in sample collection and testing and the creation and management of an appropriate data management system. In addition to fixed investments, there would also be additional per-sample costs like consumables and technician time. This would require an appropriate legislative framework and enforcement that is currently missing. If appropriately resourced, a surveillance program such as this would aid in generation of evidence and better adherence of withdrawal periods hence protecting the health of pork consumers from harmful effects of antibiotic residues.

In regulating antibiotic usage, regulatory authorities should encourage farmers to engage with cost-effective disease prevention alternatives. These include timely vaccinations, emphasis on biosecurity measures at the farm and appropriate stocking density. These alternatives will support animal health by reducing both the transmission and susceptibility to pathogens, thus reducing overreliance of antibiotics. These lessons should be taught using agricultural economic models that illustrate the benefits of disease prevention over treatment. Thereafter, legislation may be amended to require testing for residues in locally consumed meat. Such recommendations will supplement Kenya’s National Action Plan of containing antimicrobial resistance and maintaining the efficacy of antimicrobials. 

Very few willingness-to-pay studies have been performed for animal source products in Kenya, though one recent study indicates that antibiotic use had a negative impact on consumers’ willingness-to-pay for chicken [29]. Similar results have been found in many other countries indicating that it is likely that consumers, particularly more affluent urban consumers, may be less willing to purchase meat with residues. This consumer pressure has the potential to drive future legislative development in Kenya and should be further explored. 

At this stage, we have not identified any gross pathological lesions that may be used as predictors for the presence of antibiotic residues that could be utilised as an indicator in a risk-based surveillance program. Future studies should examine antimicrobial use at the farm level and may find other relevant factors which may be useful indicators for risk-based surveillance.

## 4. Materials and Methods

### 4.1. Ethical Approval 

This study was approved by the International Livestock Research Institute, Institutional Animal Care and Use Committee (ILRI IACUC Ref no. 2019-36) and the Institutional Research Ethics Committee (ILRI-IREC 2020-14). An additional permit was obtained from the National Commission for Science, Technology, and Innovation permit (NACOSTI/P/20/4847). Both the national and county Directorate of Veterinary Services granted permission to conduct the study at the local abattoir.

### 4.2. Study Site and Data Collection Procedure

This cross-sectional study was conducted between 5 January and 5 March 2021 in a medium-sized, non-integrated abattoir, which slaughters an average of 215 pigs per week for sale. Consumption occurs in Nairobi and its environs [30]. A minimum sample size for estimation of antibiotic residue prevalence was calculated as 384 based upon a 5% level of precision, 95% confidence interval and an assumed prevalence of 50% due to lack of previous data in Kenya [31]. 

This antibiotic residue study was embedded in a larger study, researching food safety and animal welfare themes that required 529 pigs to be sampled. The study site and sampling strategy have previously been described in detail [20]. All pigs brought to the abattoir were eligible for sampling. A systematic sampling method was used where the first person presenting a pig to slaughter after 6 a.m. was the first to be recruited on each sampling day. After that, every second pig presented to slaughter was recruited into the study. If a presented pig had originated from the same farm as the previously recruited pig (i.e., belonged to the same batch), this pig would be skipped and the next pig from a separate batch would be recruited to reduce the impact of clustering on our prevalence results. 

We had a team of 7 members and for ease of data collection the abattoir was divided into 3 stations, i.e., recruitment, evisceration area and dispatch point with each member assigned different roles. 

At the recruitment station, the person presenting the pig for slaughter was approached and study objectives explained to him or her. If they agreed to participate, an informed consent form would be filled in, and data collection begun. Data on the pig’s origin, farm size, husbandry type, sex and live weights were collected and recorded. We did not consider breed since the majority of the pigs presented to the slaughterhouse were of ‘European origin or European mix’ with previous studies on pig breeds in Kenya showing a heterogeneous control [32]. Therefore, identifying the breed that matches the pig’s phenotype and genotype would have been difficult or even impossible. All recruited pigs were stunned and exsanguinated by the slaughterhouse workers. An ear tag was applied on each recruited pig for ease of follow-up along the slaughter line. 

At the evisceration area, carcasses were visually examined for any gross pathological lesions. Two members of the research team, both qualified veterinarians, were responsible for inspecting the carcasses and collecting biological samples. One member examined the carcass for external lesions including ear marks (lacerations made on the ears of pigs with a sharp object to identify their pigs), tail bites, loin bruising, tether wounds, hindlimb bursitis and lacerations. The presence and absence and severity of each lesion was recorded as 1 and 0, respectively. The second team member examined thoracic and abdominal cavities and associated visceral organs namely lungs, heart and liver for cysts, abscesses, pleurisy, pneumonia, and milk spots in the liver. The lungs, liver and heart were collected, labelled and put in Ziploc bags for further examination at ILRI post-mortem room for gross pathological lesions. Finally, at the dispatch point, a 9 cm by 7 cm by 2 cm sample was collected from the left-side of *Biceps femoris* muscle [20]. These samples were placed in prelabelled Ziplock bags and kept in a cool box (at approximately 4 °C) for transport to ILRI laboratories within 2 h after sampling. Meat samples were taken from every available carcass until the sample size for the antimicrobial residue prevalence study was obtained. In the laboratory, the samples were frozen at −80 °C for later testing for antibiotic residues. The samples were stored for three months and tested between 11 June and 24 June 2021 once consumables had been delivered to Kenya. 

In the post-mortem room, lungs, heart and liver samples were inspected for lesions that may have been missed due to the rapid slaughter process at the abattoir. The 7 lung lobes were scored according to the BPEX Pig Health Scheme [33]. The cranial and caudal lobes on the left and right side were scored between 0 and 10 while the accessory lobe and the two middle lobes were scored between 0–5 depending on their health condition. The maximum score was 55 and the minimum score was 0 denoting a healthy lung [33].

All collected data were entered into an Open Data Kit (ODK) form on a mobile phone (https://opendatakit.org/) and later uploaded to the ILRI server. Data was cleaned and checked for consistency on a weekly basis. Each sampling day took an average of 4 h.

### 4.3. Antibiotic Residue Testing 

Pork samples were tested for antibiotic residues using the Premi^®^Test kit (R-Biopharm AG, Germany). This kit is a micro-inhibition screening test containing *Bacillus stearothermophilus* spores in an agar medium and a bromocresol purple indicator. In the absence of inhibitory substances, antibiotics in this case, the spores germinate and multiply to form an acid. This causes the bromocresol indicator to change colour from purple to yellow. If antibiotics are present above the MRL, the colour of the indicator remains purple and was recorded as a positive result [34]. Premi^®^Test is one of the most commonly used screening test for β-lactams, macrolides, tetracyclines and sulphonamides in meat in the EU with confirmatory tests generally utilised to confirm positive samples [22]. 

Before testing the 387 pork samples, we ran positive and negative controls. The negative meat sample was obtained from Farmers Choice. This a model firm and food processing plant that adheres to best farming practices, withdrawal periods and food safety standards. Its design and operations have been certified under ISO 22000:2005 on Food Safety Management Systems. The facility is licensed by the Director of Veterinary Services to exports their meat products and we are confident that residues would not be present. We bought 500 g of pork from Farmers Choice and obtained 100 µL of meat juice using a meat juice extractor. Further, 20 µL of meat juice was added to 2 ampoules. In addition, 2 mL of Betamox^®^ (Norbrook, UK), which contains Penicillin as its active ingredient, was spiked into the positive control ampoule. Nothing was added to the negative control ampoule. The Premi^®^Test incubator was pre-heated to 64 ºC. The 2 control ampoules were placed in the incubator, covered with foil, and incubated for approximately three hours. After this period, the negative control turned yellow while the positive control retained it purple colour as shown in Figure 1. 

The frozen pork samples were defrosted on the bench, and a meat juice extractor was used to extract 100 µL of meat juice from each of the defrosted pork sample. The juice was vortexed to obtain a homogenous sample. Further, 20 µL of the vortexed juice was added to a pre-labelled ampoule corresponding to the pig ID and incubated at room temperature for 20 min. The ampoules were turned upside down to pour out meat juices and flushed with deionised water. They were then placed on a rack in an upside down position for 5 min to allow them to dry completely.

The ampoules were then placed in the incubator that had been preheated to 64 ºC, as seen in Figure 2. The ampoules were covered with foil and incubated for approximately 3 h until the negative control turned yellow. The resultant colour for each ampoule was read against the provided colour chart and the result recorded in the ODK tool for later analysis. A negative control was always included in every batch of tested samples. This served as a guide to when to stop the incubation.

### 4.4. Data Analysis

All the uploaded data were downloaded as .csv files (comma-separated values files) from the ILRI server, cleaned and loaded into R version 4.2.2 for analysis (https://www.R-project.org/). Prevalence of antibiotic residues at 95% confidence interval was calculated using the epi.prev function under the epiR package [35]. This package accounted for the diagnostic performance of the test kit which has a reported 72.5% sensitivity and 95.5% specificity [36]. Univariate analysis was performed to explore the association between each potential predictor variable and the presence of antibiotic residues using the arsenal package in R [37]. 

A logistic regression model was then built utilising the generalised linear model function in the MASS package, utilising the binomial family since the outcome was binary [38]. An initial model was built with variables demonstrating potential association in univariate analysis with *p* < 0.2 as shown in Table 1. The best model was determined by looking at the variable combinations with the lowest AIC using the dredge function from MuMin package [38]. We tested for multicollinearity among independent variables by regressing these variables against each other. From the R^2^ output, we calculated the variance inflation factor using the formula VIF_j_ = 1/(1 − R^2^). Since they were all below 2, they were all retained in the model. 

## 5. Conclusions

Close to half of the pork destined for consumption in Nairobi and its environs contained antibiotic residues above the recommended maximum residue limit. This poses potential public health risks to pork consumers. Action is urgently needed to address the underlying antibiotic misuse that has resulted in such a high prevalence of residues in pork. We recommend that regulating authorities strongly consider routine testing of carcasses for antibiotic residues for the protection of public health. We acknowledge that the establishment of a residue surveillance program would require huge investment in diagnostic technologies, e.g., high-performance liquid chromatography mass spectrometry for confirmatory testing, training of staff in sample collection and testing and the creation and management of an appropriate data management system. In addition to the fixed investments, there would be additional per-sample costs (e.g., consumables, technician time). To reduce the cost of residue surveillance, a sampling plan would be required, which could be based upon random or risk-based sampling. 

The lack of a significant association between gross pathological lesions and the presence of antibiotic residues suggests that, for the time being, we cannot recommend a risk-based surveillance system based on gross lesions as predictors of residues in this population until antibiotic access enforcement is addressed. Future studies should research farming practices and antimicrobial use contributing to the high prevalence of residues.

To ensure better practices in the sale and use of antibiotics, appropriate legislation must be developed and enforced. There needs to be random testing of meat samples for residues and condemning of carcasses that test positive for residues. Meanwhile, producers should be sensitized on the judicious use of antibiotics and importance of adhering to recommended withdrawal times. Antibiotic stewardship, food safety and improved health outcomes will be ensured through this multifaceted approach.

## Figures and Tables

**Figure 1 antibiotics-12-00492-f001:**
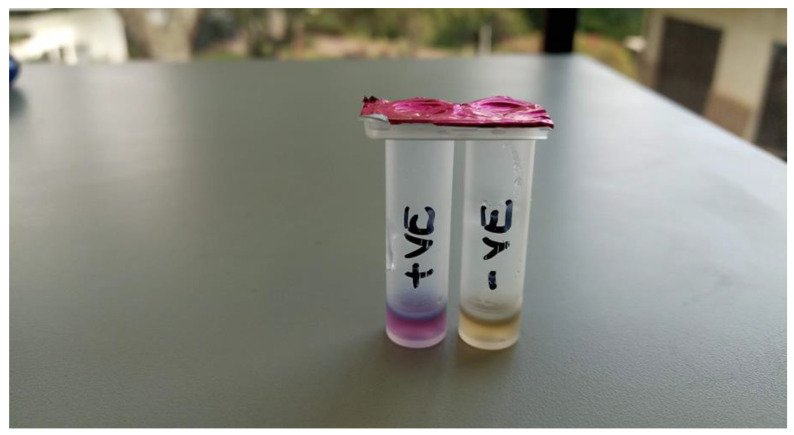
Images of positive and negative controls after incubation.

**Figure 2 antibiotics-12-00492-f002:**
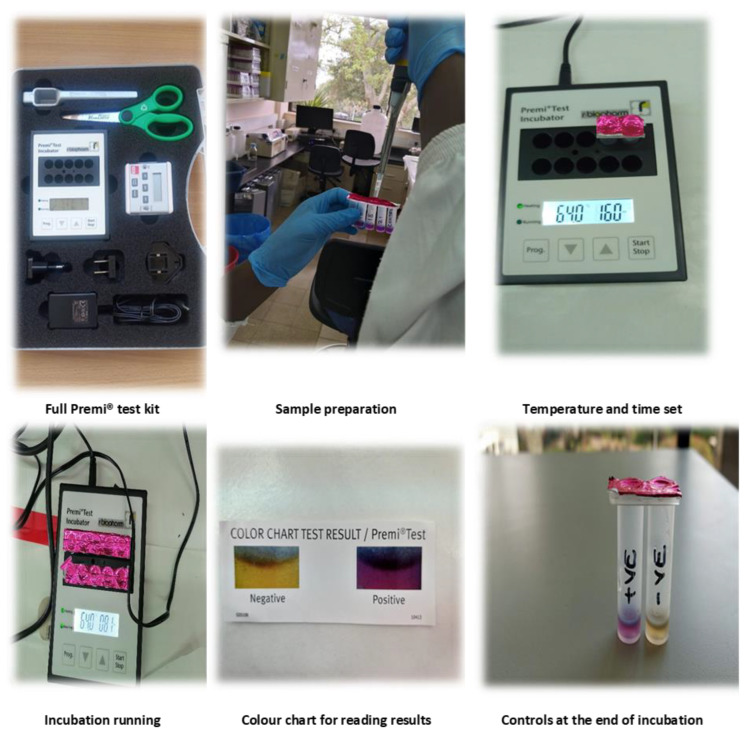
Premi^®^Test kit and summary of incubation process.

**Table 1 antibiotics-12-00492-t001:** Univariate analysis results showing the association between the presence of antibiotic residues over the recommended MRL and different predictor variables. The level of association is denoted by the *p* value in the last column.

Variables	N *	Variable Observation	Residue Result		*p* Value
			Negative	Positive	
Pleuropneumonia	266				**0.727**
		Absent	125	61	
		Present	52	28	
Tail bites	364				**0.107**
		Absent	233	116	
		Present	13	2	
Liver milk spots	348				**0.612**
		Absent	220	107	
		Present	13	8	
Loin bruising	364				**0.423**
		Absent	223	114	
		Present	13	4	
Hind limb bursitis	364				**0.111**
		Absent	241	112	
		Present	5	6	
Tether lesions	364				**0.868**
		Absent	239	115	
		Present	7	3	
Lung abscess	266				**0.480**
		Absent	175	87	
		Present	2	2	
Lacerations	364				**0.961**
		Absent	242	116	
		Present	4	2	
Cysts in the liver	362				**0.480**
		Absent	243	118	
		Present	1	0	
Pleurisy	266				**0.593**
		Absent	173	86	
		Present	4	3	
Husbandry type	381				**0.558**
		Housed	247	122	
		Outdoor	9	3	
Sex	384				**0.084**
		Female	143	58	
		Male	115	68	
Farm size	381				**0.72**
		<10	167	80	
		10 < 50	59	34	
		50 < 100	2	1	
		>100	28	10	
Lung score	266				**0.642**
		Mean (SD)	6.96 (13.40)	7.78 (14.21)	
		Range	0.00–55.00	0.00–48.00	
Live weight	366				**0.199**
		Mean (SD)	60.996 (28.308)	57.193 (22.257)	
		Range	13.0–230.0	27.0–157.0	
County of origin	383				**0.558**
		Homabay	1	0	
		Kajiado	6	3	
		Kiambu	200	103	
		Makueni	0	0	
		Murang’a	1	1	
		Nairobi	39	16	
		Nakuru	11	2	

* N varies based on the data that was successfully collected. Data was unavailable on all variables due to the rapid slaughter process and some traders being unwilling or unable to provide data, or to allow us to purchase the biological samples for closer inspection at the post-mortem room.

**Table 2 antibiotics-12-00492-t002:** Logistic regression output for the best model.

Variable	Estimate	Standard Error	Z Value	*p* Value
Intercept	−0.9020	0.1644	−5.487	4.1 × 10^−8^
Sex male	0.3761	0.2276	1.652	0.0985
Presence of tail bite	−1.292	0.7806	−1.655	0.0979
Presence of hindlimb bursitis	1.0651	0.6278	1.697	0.0898

## Data Availability

All data is freely available at: https://doi.org/10.17638/datacat.liverpool.ac.uk/1441.

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
