# Peer review of "Prevalence of Antibiotic Residues in Pork in Kenya and the Potential of Using Gross Pathological Lesions as a Risk-Based Approach to Predict Residues in Meat"

_antibiotics, 2023, doi:10.3390/antibiotics12030492_

Round 1
Reviewer 1 Report
The article is interesting. Only a point of revision:The used analytical method is a screening method. It doesn't have the same ccbeta for all molecules and it doesn't take account of the presence of more tha antibiotics (perhaps all lower than LMR). You should explain this because it is impossible to confirm the levels over the LMR without a confirmatory method (spectrometric tecniques). I can understand the difficulties in that country but this shall be remarked in the article. Different way if the use of this method is allowed for the declaration of conformity (or not) in Kenya.
Author Response
Reviewer 1
The article is interesting. Only a point of revision: The used analytical method is a screening method. It doesn't have the same ccbeta for all molecules and it doesn't take account of the presence of more tha antibiotics (perhaps all lower than LMR). You should explain this because it is impossible to confirm the levels over the LMR without a confirmatory method (spectrometric techniques). I can understand the difficulties in that country but this shall be remarked in the article. Different way if the use of this method is allowed for the declaration of conformity (or not) in Kenya.
Thank you for this observation. Based on the funds available, we were only able to perform a screening test using Premi® test. We have included a paragraph on this limitation in the Discussion (lines 152-160) and made reference to the need for confirmatory tests in the Methods section (lines 337 -339)
Reviewer 2 Report
The study has merit and relevance, especially in situations where regulatory frameworks, and monitoring is limited. Please state if ethical clearance was obtained and what it entailed. Which antibiotics are commonly used in this farming system and for what purposes (treatment or prevention of diseases)? How were the negative and positive controls decided? Please see document for comments.

Author Response
Reviewer 2
The study has merit and relevance, especially in situations where regulatory frameworks, and monitoring is limited. Please state if ethical clearance was obtained and what it entailed.
Our ethical clearance statement has now been moved to the methods sections (lines 247 - 254) to make it clear for the readers.
Which antibiotics are commonly used in this farming system and for what purposes (treatment or prevention of diseases)?
Tetracyclines, sulphonamides, aminoglycosides and beta-lactams are the most commonly used classes of antibiotics in treatment and prevention of infections in food animals in Kenya with no current evidence of their use specifically for growth promotion. This information has been included in the manuscript and source cited (lines 55 – 57).
How were the negative and positive controls decided?
We have included the following explanation of the positive and negative controls in lines 341-356
“Before testing the 387 pork samples, we ran positive and negative controls. Negative meat sample was obtained from Farmers Choice. This a model firm and food processing plant that adheres to best farming practices, withdrawal periods and food safety standards. Its design and operations have been certified under ISO 22000:2005 on Food Safety Management Systems. The facility is licensed by the Director of Veterinary Services to exports their meat products and we are confident that residues would not be present. We bought 500 grams of pork from Farmers Choice and obtained 100 µL of meat juice using a meat juice extractor. 20 µL of meat juice was added to two ampoules. Two millilitres of Betamox® which contains Penicillin as its active ingredient was spiked into the positive control ampoule. Nothing was added to the negative control ampoule. The Premi®Test incubator was heated to 64℃. The two control ampoules were placed in the incubator, covered with foil, and incubated for approximately 3 hours. After this period the negative control turned yellow while the positive control retained it purple colour as shown in figure 1. “
COMMENTS WITHIN THE MANUSCRIPT.
Abstract
Are antibiotics in the feed or administered separately for treatment of diseases?
Tetracyclines, sulphonamides, aminoglycosides and beta-lactams are the most commonly used classes of antibiotics in treatment and prevention of infections in food animals in Kenya with no current evidence of their use specifically for growth promotion. This information has been included in the manuscript and source cited (lines 55 – 57).
Results
Would it also not be due to not wishing to know if their carcasses have acceptable antibiotic residual levels, as this directly impacts their sales and may force them to change some husbandry aspects?
We have indicated (table 1) that traders were unwilling or unable to provide some data (including meat samples). We do not know the reasons for this and you are right some maybe that they do not wish to know the residue status of their animals, although we made it clear that results would be anonymized and aggregated before they were reported. We have included in the discussion a note on consumers willingness to pay as a potential driver of legislative change (lines 231-238)
Discussion
This observation was from using the meat juice after the samples were frozen and then thawed, would the result be similar if the meat was cooked?
Whilst some studies have suggested that cooking meat products may reduce antibiotic concentration, residue testing for surveillance was performed in uncooked products. We are advocating for solving the problem at the source which is at the farm level through adhering to recommended withdrawal periods and antibiotic stewardship by animal health practitioners. In the long run, we hope that lessons from European Union member countries on banning banned antibiotics as growth promoters will be adopted. These interventions will lead to a reduction in antibiotic residues in meat products, protecting public health.
This conclusion is not aligned with the study title, which is definitive, suggesting that there is indeed prevalence of antibiotic residues in pig carcasses. Please revise the title to suggest an inquiry into this perception.
Our conclusion aligns with the study title as we determined the prevalence of residues in pork which we indicate here was close to half. We also aimed to determine if any gross lesions in this population were associated with the presence of residues and if they could be utilised as indicators in a risk-based system but found this was not the case. We, therefore, consider our title and conclusion to be aligned.
Methods
Was a survey or inquiry conducted on these farms, individually, to deduce their use of antibiotics in their disease management strategies? What type of antibiotics are prevalently used and dosages? Were there any farm records to review?
Please provide a clear and detailed experimental design. However, if it does not apply in the current study, be specific in describing the actual criteria that was used.
This study was conducted at a slaughterhouse (see lines 257 -259), not on farms so we were unable to determine individual farmers antimicrobial use. Most of the people presenting the pigs for slaughter were pig traders who are unable to provide history of antibiotic use. With the high prevalence rates of residues, we recommend that future studies on antimicrobial use be conducted at the farm level line and have recently undertaken an on-farm study on the same (In Prep).
We have provided enhanced detail on the study methodology in lines 256-327
Why was breed type not considered?
We didn’t consider breed since the majority of the pigs presented to the slaughterhouse are of a ‘European or European mix’ type pig, and previous studies on pig breeds in Kenya have shown a heterogeneous admixture and observation of the phenotype may not accurately indicate the pigs’ genotype so accurate differentiation of breed type would be impossible. This has been explained in lines 285-288
What was in the negative control? Was the negative control deduced from L233-239 or from pigs that had not been treated with antibiotics?
As indicated above we have included the following explanation of the positive and negative controls in lines 341-356 .
Reviewer 3 Report
See attached peer review.

Author Response
“The manuscript is organized such that the Results section follows the Introduction and the Discussion follows the Results. Then, Materials and Methods is presented before the Conclusions section. Obviously, this is not the standard organization for
a manuscript. However, it can be acceptable, providing the Editorial Staff of the Journal find it
acceptable
We acknowledge that the order seems a little unorthodox, but we followed the structure and template laid out in the ‘Instructions to Authors’ from the journal as below. We would be happy to move the Materials and Methods before the results section if the editor agrees.
Research manuscripts should comprise:
- Front matter: Title, Author list, Affiliations, Abstract, Keywords.
- Research manuscript sections: Introduction, Results, Discussion, Materials and Methods, Conclusions (optional).
- Back matter: Supplementary Materials, Acknowledgments, Author Contributions, Conflicts of Interest, References.
Introduction, p. 2, ll. 45-47: In this sentence, the authors state that coping with the increased demand for animal-sourced protein will lead to livestock keepers using more antibiotics to prevent diseases and promote growth. The authors should note that there are major efforts in North America and Europe to reduce use of antibiotics in livestock production. Thus, this sentence may not be true for all areas. The authors should acknowledge this, or possibly focus this sentence on Africa or “less developed countries.”
We have added a citation stating that antimicrobial use is reducing in European countries which contrasts to low and middle-income countries where usage is expected to rise due to lack of regulation. Lines 63 -66
Introduction, p. 2, ll. 48-54: It would be appropriate to include the date these data were published. As an example, for the first sentence of the paragraph: “Globally, it was reported in 2017 that about 73% of all antimicrobials.....”
This has been added to the manuscript as suggested. Line 48
Results, p. 3, ll. 102-104: Please give an explanation of how the 529 pigs were “recruited” for the study.
We have edited the results to indicate that 387 pork samples were collected for analysis. The original 529 pigs is the total number of pigs recruited (entered into) the larger study into which this prevalence investigation was embedded as now better described in the methods sections
Results, pp. 3-4, Table 1, ll. 117-119: It is very hard for me to understand this Table and justify the numbers. For example, for pleuropneumonia, it looks like 94 carcasses had lesions out of a total of 344 tested. However, under total of 387 is reported with 261 negative and 126 positive. Please be sure that this explained to the reader make it very clear what the numbers refer to. I am not sure why pleuropneumonia has an N of 344 and tail bites have an N of 484.
Although 529 pigs were entered into the full study, the slaughter process was such that the team was unable to gather all data points from each carcass. It was also the case that after consenting to the study some traders were in a rush to leave the abattoir and left before samples could be taken, or they withdrew from the study when we requested to obtain the pluck or meat samples. We also ceased to collect This resulted in a variable N for different lesions depending on the availability of the relevant data. The total (N) for each univariate analysis in Table 1 has been made clear in column 2 as well as the table caption.
We note as we had previously merged 2 results tables including all pigs in the wider study this made it difficult to interpret the results. The table has now been adjusted to show data only from the subset of 387 pigs with pork samples available for residue testing.
Results, p. 5, ll. 128-130: Verb needs to be changed: “The variance inflation factors output for .......was1.005, 1.008, ..................” The WAS should be changed to WERE. Please add commas to this portion: “.....were 1.005, 1.008 and 1.01, respectively, indicating .............”
These changes have been corrected as suggested.
Results, p. 5, ll. 133-134, Table 2: Perhaps I don’t understand, but is “P Value” for “Intercepts” correctly as “4.1e-08”?
That is right. That is the p-value for the intercept.
Results, p. 6, l. 174: Is this meant to be a new paragraph and therefore should be indented?
The statement is meant for that current paragraph. It shows the effect of having a National Action Plan in place.
Results, p. 7, l. 215: The term “silverside” is not a term used universally and can mean different things in different countries. Could the authors use another term or explain the term.
The term ‘silverside’ has been deleted and now reads as ‘left side of Bicep femoris muscle’.
Results, p. 7, l. 222: When you refer to “swift” do you refer to the speed of the slaughter process? If so, would “rapid” be a better term to describe it?
We have replaced ‘swift’ with ‘rapid’ to indicate the speed of the slaughter process.
Results, p. 7, l. 224: Please provide a reference or website for BPEX Pig Health Scheme.
This reference has been provided.
Results, p. 7, l. 233: Please provide a source for the Premi® test kit.
The Premitest kit was sourced from R-Biopharm AG, Germany and this has been added on discussion section line 152 and methods line 331
Results, p. 7, ll. 218-220: Could you make any statement relative to whether the antibiotics you are testing can be frozen and not affect concentration?
We have included the following passage in the discussion (lines 170 -177)
“Freezing has been demonstrated to reduce the concentration of some antibiotics in meat, which may have resulted in an artificially reduced prevalence of samples with residues above the MRL in our study. The effect does, however, appear to be time related and we therefore expect that immediate freezing of the samples at -80°C for under six months with a single defrosting event will have reduced the impact of freezing to an acceptable level. More research is required to quantify what impact this may have had and is highly relevant for the design of future studies or proposed surveillance activities.”
Conclusions, p. 9, ll. 277-296: Would you want to make any comment about the cost and time required to institute such a program?
We have included the following in the manuscript lines 209 - 214 “The establishment of such a residue surveillance program would require investment in diagnostic technologies (eg mass spectrometry for confirmatory testing), training of staff in sample collection and testing, and the creation and management of an appropriate data management system. In addition to the fixed investments there would be additional per-sample costs (consumables, technician time). It would also require an appropriate legislative framework and enforcement which is currently missing.”
We do not wish to comment on specific costs as we have not undertaken specific data collection to confirm the relevant associated costs.
References:
Are reference #’s 11, 13, 23 and 25 complete?
We checked the references highlighted in the earlier version and we confirm that they are now complete. The mentioned references numbers [11,13, 23 and 25] may have changed in the latest version of the manuscript. They are now listed with reference numbers [13, 15, 35, 37]
Reviewer 4 Report
Very novel, well-designed, and well-written.
No required edits for publication.
Author Response
No required edits for publication.
Thank you.